# In Silico and In Vivo Evaluation of the Maqui Berry (*Aristotelia chilensis* (Mol.) Stuntz) on Biochemical Parameters and Oxidative Stress Markers in a Metabolic Syndrome Model

**DOI:** 10.3390/metabo13121189

**Published:** 2023-12-07

**Authors:** Emily Leonela Castillo-García, Ana Lizzet Cossio-Ramírez, Óscar Arturo Córdoba-Méndez, Marco A. Loza-Mejía, Juan Rodrigo Salazar, Edwin Chávez-Gutiérrez, Guadalupe Bautista-Poblet, Nadia Tzayaka Castillo-Mendieta, Diego A. Moreno, Cristina García-Viguera, Rodolfo Pinto-Almazán, Julio César Almanza-Pérez, Juan Manuel Gallardo, Christian Guerra-Araiza

**Affiliations:** 1Unidad de Investigación Médica en Farmacología, Hospital de Especialidades Dr. Bernardo Sepúlveda, Centro Médico Nacional Siglo XXI, Instituto Mexicano del Seguro Social, Mexico City 06720, Mexico; cbs2202800924@tlitani.uam.mx (E.L.C.-G.); cbs2221801021@xanum.uam.mx (G.B.-P.); 2Doctorado en Ciencias Biológicas y de la Salud, Universidad Autónoma Metropolitana, Mexico City 52919, Mexico; 3Maestría en Ciencias de la Salud, Sección de Estudios de Posgrado e Investigación, Escuela Superior de Medicina, Instituto Politécnico Nacional, Mexico City 11340, Mexico; acossior2100@alumno.ipn.mx; 4Design, Isolation, and Synthesis of Bioactive Molecules Research Group, Chemical Sciences School, Universidad La Salle-México, Benjamín Franklin 45, Mexico City 06140, Mexico; oc@lasallistas.org.mx (Ó.A.C.-M.); marcoantonio.loza@lasalle.mx (M.A.L.-M.); juan.salazar@lasalle.mx (J.R.S.); 5Doctorado en Ciencias en Biomedicina y Biotecnología Molecular, Escuela Nacional de Ciencias Biológicas, Instituto Politécnico Nacional, Prolongación Manuel Carpio y Plan de Ayala s/n, Mexico City 11340, Mexico; echavezg1700@alumno.ipn.mx; 6Postdoctorate-Conacyt-Unidad de Investigación Médica en Enfermedades Neurologicas, Hospital de Especialidades, Centro Médico Nacional Siglo XXI, Instituto Mexicano del Seguro Social, Av. Cuauhtémoc 330 Col. Doctores, Mexico City 06725, Mexico; tzayakita@gmail.com; 7Laboratorio de Fitoquímica y Alimentos Saludables (LabFAS), CEBAS, CSIC. Campus Universitario de Espinardo-25, E-30100 Murcia, Spain; dmoreno@cebas.csic.es (D.A.M.); cgviguera@cebas.csic.es (C.G.-V.); 8Sección de Estudios de Posgrado e Investigación, Escuela Superior de Medicina, Instituto Politécnico Nacional, Plan de San Luis y Díaz Mirón, Mexico City 11340, Mexico; 9Laboratorio de Farmacologia, Departamento de Ciencias de la Salud, DCBS, UAM-I, Mexico City 09310, Mexico; jcap@xanum.uam.mx; 10Unidad de Investigación Médica en Enfermedades Nefrológicas, Hospital de Especialidades, Centro Médico Nacional Siglo XXI, Instituto Mexicano del Seguro Social, Mexico City 06720, Mexico; jmgallardom@gmail.com

**Keywords:** metabolic syndrome, maqui berry, molecular docking, diet-induced model, oxidative stress

## Abstract

Metabolic syndrome (MetS) is a complex disease that includes metabolic and physiological alterations in various organs such as the heart, pancreas, liver, and brain. Reports indicate that blackberry consumption, such as maqui berry, has a beneficial effect on chronic diseases such as cardiovascular disease, obesity, and diabetes. In the present study, in vivo and in silico studies have been performed to evaluate the molecular mechanisms implied to improve the metabolic parameters of MetS. Fourteen-day administration of maqui berry reduces weight gain, blood fasting glucose, total blood cholesterol, triacylglycerides, insulin resistance, and blood pressure impairment in the diet-induced MetS model in male and female rats. In addition, in the serum of male and female rats, the administration of maqui berry (MB) improved the concentration of MDA, the activity of SOD, and the formation of carbonyls in the group subjected to the diet-induced MetS model. In silico studies revealed that delphinidin and its glycosylated derivatives could be ligands of some metabolic targets such as α-glucosidase, PPAR-α, and PPAR-γ, which are related to MetS parameters. The experimental results obtained in the study suggest that even at low systemic concentrations, anthocyanin glycosides and aglycones could simultaneously act on different targets related to MetS. Therefore, these molecules could be used as coadjuvants in pharmacological interventions or as templates for designing new multitarget molecules to manage patients with MetS.

## 1. Introduction

Metabolic syndrome is defined as a pathological condition known for abdominal obesity, insulin resistance, hypertension, and hyperlipidemia. Metabolic syndrome is a complex pathophysiological state caused mainly by an imbalance in caloric intake and energy expenditure. Still, it is also affected by the genetic/epigenetic composition of the individual, the predominance of a sedentary lifestyle over physical activity, and other factors such as the quality and composition of food [1]. Physical activity and exercise are components for preventing metabolic syndrome and go beyond the immediate benefit of caloric expenditure [2]. On the other hand, studies provided evidence supporting the beneficial role of the traditional Mediterranean diet in preventing metabolic syndrome. Some studies indicate the beneficial effects of blackberry consumption on health, particularly in chronic diseases, such as cardiovascular disease, obesity, and diabetes. However, these studies have been conducted with commercial products from strawberries, blackberries, or blueberries [3].

Another promising berry is *Aristotelia chilensis* (Mol.) Stuntz, a native Chilean berry named ‘maqui,’ consumed as a natural antioxidant, and also used to treat several diseases such as sore throat, fever, inflammation, injuries and scars, and migraines, among others [4,5,6].

It has been reported that maqui berry (MB) has a high antioxidant power due to its high content of anthocyanins compared to other berries [7,8]. The MB powder is rich in anthocyanins (>600 mg/100 g d.w. by HPLC analysis, mainly delphinidin and cyanidin glycosylated derivatives) [9], and other compounds of interest, such as ellagic acid derivatives (>14 mg/100 g d.w.), flavonols (>24 mg/100 g d.w.), and chlorogenic acid (>9 mg/100 g d.w.) [10].

The MB juice induces anti-atherogenic activity due to inhibiting the lipoprotein oxidation of low density (LDL). It protects endothelial cells against intracellular oxidative stress (OS) [5]. Also, it induces browning in the subcutaneous white adipose tissue and ameliorates insulin resistance in high-fat diet-induced obese mice [11]. On the other hand, MB extracts suppressed glucose production. They attenuated the downregulation of gluconeogenic enzyme and glucose 6-phosphatase [12], together with the inhibition of α-glucosidase activity, reducing fasting and postprandial glycemia, and insulinemia and significantly improving the serum lipid profile in prediabetic individuals [13]. Delphinidin has been reported to reduce fasting blood glucose levels in the obese C57BL/6J mouse [12]. It has also been reported that consuming cyanidin and delphinidin decreases obesity, dyslipidemia, and insulin resistance induced by a high-fat diet [14]. However, even with the previously mentioned studies showing the beneficial effects on carbohydrate metabolism and hyperglycemia, there is no information on the effects of MB on the other MetS components. Due to this, the objective of this study was to determine if the MB components can have activity on key receptors associated with blood pressure, hypertriglyceridemia, hypercholesterolemia, and hyperglycemia, as well as the evaluation of the effects of a maqui berry freeze-dried on a standardized murine model of MetS.

Due to the aforementioned, it could be considered that this is a promising berry to prevent or to help in the management of MetS.

## 2. Materials and Methods

### 2.1. MB Powder

For conducting the experiments, we utilized a maqui berry powder known as CHI2, which was obtained from the Ecuadorian Rainforest, LLC, Clifton, NJ, USA. This powder is derived from a whole fruit that is ground to be rich in anthocyanins (600 mg/100 g d.w. as per HPLC analysis). This fruit also contains other beneficial compounds such as ellagic acid derivatives (14 mg/100 g d.w.), flavonols (24 mg/100 g d.w.), and chlorogenic acid (9 mg/100 g d.w.) [15]. Glycosides and di-glycosides of delphinidin and cyanidin were recognized as the principal anthocyanidins found in MB. In previous works, Delphinidin and its derivatives Delphinidin 3-O-glucoside, Delphinidin 3,5-O-diglucoside, Delphinidin 3-O-sambubioside, and Delphinidin 3-O-sambubioside-5-O-glucoside, together with Cyanidin, and its derivatives Cyanidin 3-O-glucoside, Cyanidin 3,5-O-diglucoside, Cyanidin 3-O-sambubioside, and Cyanidin 3-O-sambubioside-5-O-glucoside, were found as its principal components of the water extract of maqui fruits [10] (Figure 1).

### 2.2. Animals

Eighty Sprague–Dawley adult rats (40 males and 40 females, 250–300 g) from the animal facility of the Centro Medico Nacional Siglo XXI, Instituto Mexicano del Seguro Social, were housed in acrylic boxes (five per cage) with controlled temperature (varying between 20 and 26 °C) and humidity conditions (range from 40 to 60%), light/dark cycles of 12/12 h (10:00 a.m. to 10:00 p.m.), and water and food (Purina LabDiet^®^ 5008 Richmond, IN, USA) ad libitum. Animals were handled according to the National Institutes of Health Guide’s guidelines and requirements for the Care and Use of Laboratory Animals (NIH Publication No. 85-23, revised 1985) and the Mexican Standard NOM-062-ZOO-1999 [16] for the production, care, and use of laboratory animals, and the Institutional National Scientific Research Committee, IMSS (approval number R-2017-755-060). All efforts were made to minimize animal suffering, and the protocol was designed to keep the number of animals used to a minimum. The rats were randomly assigned to each treatment group (n = 10 rats per sex).

### 2.3. Metabolic Syndrome Murine Model

The model used in this study was made as previously standardized and reported [17]. Animals were assigned to either continue consuming regular Chow commercial food (control groups) (n = 40, 20 males and 20 females) (Purina-Rodent Laboratory Chow-5001 3.310 kcal/g) or fed with high fructose and high-fat [standard food (60%), fructose (30%), and pork fat (10%)] Diet [HFHF diet groups (4.161 kcal/g)] for 12 weeks (n = 40, 20 Males and 20 Females), to induce MetS [17].

The diet was produced based on standard food (60%), fructose (30%), and pork fat (10%), and it was provided for 12 weeks. To obtain 1 kg of food rich in fructose and fat, 600 g of standard food (LabDiet^®^) was pulverized and mixed with 300 g of fructose and 100 g of lard until a homogeneous mixture was obtained. The mixture was cut in a manner similar to the standard diet (4161 kcal/kg).

### 2.4. Treatments

After the 12 weeks of feeding, male and female rats were randomly divided into four experimental groups as follows: (1) Control: normal diet + vehicle (water) (CM and CF), (2) Control+MB normal diet + 100 mg/kg of MB lyophilized powder diluted in water (CM+MB and CF+MB), (3) hypercaloric Diet (a diet rich in fructose 30% + fat 10%) + vehicle (HFHF-M and HFHF-F), and (4) hypercaloric diet + 100 mg/kg of MB lyophilized powder diluted in water (HFHF-M+MB and HFHF-F+MB) (n = 20 rats per group, ten male and ten females). A two-week treatment of MB or vehicle was administered.

### 2.5. Monitoring of Weight

Animals were weighed with an electronic scale (Ohaus Specialty, Parsippany, NJ, USA). Each animal’s weight was registered at the beginning of the study and weekly until the end of each feeding period.

### 2.6. Blood Pressure

The systolic and diastolic blood pressure of the rats was monitored using a non-invasive tail-cuff system (CODA™, Kent Scientific Corporation, Torrington, CT, USA) after warming the rats for 10 min.

### 2.7. Sacrifice and Blood Sample Collection

At the end of each treatment time, animals were sacrificed 24 h later by decapitation. The blood was collected and centrifuged at 2500–3000 rpm for 15 min at 4 °C. The serum of each animal was collected and stored at −80 °C until the analysis was performed.

### 2.8. Blood Serum Analysis

The serum was thawed, and the levels of glucose, total cholesterol, and triglycerides were quantified for each animal in duplicate through commercial colorimetric kits (DiaSys Holzheim, Germany) following the manufacturer’s instructions. The concentration of glucose, triacylglycerides, and cholesterol was calculated using the following formula: [S] = (SA/StA) × [St] where [S]: concentration of the sample (mg/dL); SA: absorbance of the sample; StA: standard absorbance; * [St] concentration of the standard (200 mg/dL). The Triacylglycerides and Glucose index (TyG) were calculated according to the formula: Ln [fasting triglycerides (mg/dL) × fasting plasma glucose (mg/dL)/2] [18]

### 2.9. Determination of Malondialdehyde

Before the determination of malondialdehyde (MDA), a step consisting of deproteinizing the sample was performed to avoid interference by the sediments formed by denatured proteins to avoid pipetting errors.

The sample (200 μL) was placed in Eppendorf tubes, and 200 μL of 20% trichloroacetic acid (TCA) was added, vortexed, and incubated at 4 °C for 10 min, and centrifuged at 11,000 rpm for 3 min. The supernatant was recovered. MDA is one of the final products of polyunsaturated fatty acid peroxidation and a marker of free radical activity. At an acid pH and high temperatures, one molecule of MDA reacts with two molecules of TCA, forming pink adducts detected at 532 nm [19]. A calibration curve with eight points (0–100 μM), as well as the samples, were prepared as follows: 25 μL of either the supernatant or standard solution were mixed with 200 μL of 0.8% TCA, vortexed, and incubated at 95 °C/60 min, and lastly at 22 °C/10 min. Finally, the absorbance at 532 nm was determined for the calibration curve and the samples.

### 2.10. The Activity of Superoxide Dismutase

The superoxide dismutase (SOD) activity was measured using the reduction of water-soluble formazan dye. The SOD activity protocol was followed from the 1960 SOD determination Kit (Sigma-Aldrich, Saint Louis, MO, USA). The rate of reduction of water-soluble formazan dye is linearly related to xanthine oxidase activity and inhibited by the SOD. Therefore, the SOD activity can be quantified by measuring the decrease in the color development of the dye at 440 nm. The brain regions were homogenized in an ice-cold buffer containing 100 mM KH_2_PO_4_, 1 mM EDTA, and 0.1% triton X, pH 7.5. Three blanks were included for the assay: no sample (blank 1), a specific sample with no enzyme (blank 2), and no sample and no enzyme (blank 3). The blank 2 acts as a control for non-specific formazan reduction as it does not contain the xanthine oxidase enzyme, for which the non-specific electron donors will account for all formazan reduction. The activity was performed in a 96-well microplate. All the measurements were performed at 23 °C. The SOD activity rate was calculated with the following equation and expressed as an inhibition rate percentage.
SOD activity = ((Abs_blank1_ − Abs_blank3_) − (Abs_sample_ − Abs_blank2_)/(Abs_blank1_ − Abs_blank3_)) × 100

### 2.11. Protein Carbonyls

The total protein carbonyls were determined employing the dinitrophenylhydrazine (DNPH) method described by Levine et al. [20] with some modifications. Protein suspensions (200 μL) from each experimental unit were precipitated by the addition of 1 mL of cold 10% trichloroacetic acid (TCA), followed by centrifugation at 4 °C at 600× *g* for 5 min, and the supernatants were discarded. The pellets were treated with 1 mL of a 2 M HCl solution with 0.2% DNPH and incubated at room temperature for one hour. Proteins were precipitated with 1 mL of cold 10% TCA, followed by centrifugation at 4 °C, 1200× *g* for 10 min, and washed twice with 1 mL of ethanol: ethyl acetate (1:1 *v*/*v*). The pellets were dissolved in 1.5 mL of 20 mM Na_3_PO_4_ buffer pH 6.5 and added with guanidine hydrochloride to reach 6 M. The number of carbonyls was expressed in nanomoles of protein hydrazones per mg of protein using a molar extinction coefficient of hydrazones (21.0 nM^−1^ cm^−1^) with absorbance readings at 370 nm [21].

### 2.12. Statistical Analysis

Data were presented as the mean ± standard error (SEM). To analyze food consumption and total caloric intake of macronutrients, a three-way ANOVA analysis was used to evaluate the differences between groups. The PAT results were assessed using a Kruskal–Wallis test followed by a Dunn post hoc test. The GraphPad Prism software version 8 (Prism 8.0.2, GraphPad Software Inc, Boston, MA, USA) was used for the analyses. A *p*-value < 0.05 was considered statistically significant.

### 2.13. Molecular Docking

A previously reported procedure was used for molecular docking [22]. The maqui extract is a rich source of glycosylated derivatives of delphinidin and cyanidin (Figure 1). Therefore, the main anthocyanins previously described were constructed using Spartan’10 (Wavefunction, Inc, Irvine, CA) and their geometry was further optimized using the MMFF // ab initio 6-31G*. Then, the optimized structures were exported to Molegro Virtual Docker (Qiagen Bioinformatics, Aarhus, Denmark) [23]. The structures of selected enzymes and receptors related to MetS were retrieved from the Protein Data Bank [24] with the following PDB accession codes: CETP (PDB code: 2OBD), ACAT-2 (PDB code: 1WL4), HMGCoA reductase (PDB code: 1HWK), PPAR-α (PDB code: 1I7G), ACE (PDB code: 1O86), GPR40 (PDB code: 4PHU), DPP-IV (PDB code: 4A5S), PPAR-γ (PDB code: 1I7I), PTP1B (PDB code: 1XBO), α-glucosidase (PDB: 3TOP), and PPAR-δ (PDB code: 3GZ9).

All the water molecules and co-crystallized ligands were removed from the structure. The searching site was centered in the active site or ligand-binding domain of each enzyme/receptor and delimited by a sphere of 12 Å in diameter and a grid of 0.3 Å. A MolDock Optimizer was used as the searching algorithm using 25 runs, and the Rerank score was selected as the scoring scheme. Poses with the lowest score were used for the ligand–receptor complex analysis. The co-crystallized ligand of each structure was also docked and compared to the original pose in the downloaded structure to verify the docking procedure’s accuracy; in all cases, a root-median-square-deviation (RMSD) value lower than 2.5 Å was recorded.

### 2.14. Molecular Dynamics

Molecular dynamics (MD) simulations were performed to provide additional support to the molecular docking results and gain some information about the predicted complexes’ stability using a previously used methodology [25]. The MD simulations were carried out in the YASARA Dynamics v.18.4.24 [26] (Yasara Gmb, Vienna, Austria) using the AMBER14 force field [27]. The MD simulation’s initial structures were obtained from the docking complexes of delphinidin and its glycosylated derivatives with PPAR-α, PPAR-γ, α-glucosidase, and ACE because these were the most probable targets according to the docking results. Each of the twenty complexes was positioned in a water box with a size of 100 Å × 100 Å × 100 Å, with periodic boundary conditions (PBC). The temperature was set at 298 K, and the water density was set at 0.997 g/cm^3^. The sodium (Na^+^) and chlorine (Cl^−^) ions were included for neutralization and to provide conditions that simulate a physiological solution (pH 7,4, NaCl 0.9%). The particle–mesh Ewald (PME) algorithm with a cut-off radius of 8 Å was applied. The simulation snapshots were recorded at intervals of 250 ps with a timestep of 2.5 fs until a total simulation time of 20 ns was reached. It was considered that the complex reached equilibrium if the RMSD variations were smaller than 2 Å for at least 15 ns. The resulting MD trajectories were then analyzed with a series of macros included as part of the YASARA software, including the RMSD variations along time and ligand binding energy (LBE) using molecular mechanics energies combined with the Poisson–Boltzmann and surface area continuum solvation (MM-PBSA) calculations. The snapshots of the last ten ns were chosen for LBE calculation.

## 3. Results

### 3.1. Effect of MB Administration on Metabolic Parameters in Rats Fed HFHF Diet

To assess the effect of the MB on animals fed with an HFHF diet, parameters such as body weight, the concentration of glucose, triacylglycerides, and total cholesterol related to MetS were assessed. It was observed that after 12 weeks, the groups fed with the HFHF diet increased weight (main effect of diet exposure: F (3, 72) = 13.55, *p* < 0.0001; main effect of sex (males>females): F (1, 72) = 705.8, *p* < 0.0001) (Figure 2A); glucose (main effect of time diet exposure: F (1, 72) = 14.51, *p* < 0.001) (Figure 2B); cholesterol (main effect of time diet exposure: F (1, 72) = 18.32, *p* < 0.001) (Figure 2C); and triacylglycerides (main effect of time diet exposure: F (1, 72) = 17.77, *p* < 0.001) (Figure 2D); than their respective control group.

After a two-week MB treatment, it was observed the main effect of diet exposure [F (1, 72) = 72.56, *p* < 0.0001] with the main sex effect [F (males > females): (1, 72) = 1401, *p* < 0.0001) and interaction between diet and sex [F (1, 72) = 18.96, *p* < 0.0001) on body weight change. Nevertheless, even when the MB groups reported the main effect of the treatment [F (1, 72) = 12.58, *p* = 0.0007], no interaction between sex × MB [F (1, 72) = 0.0263, *p* = 0.8715] nor diet × MB [F (1, 72) = 1.805, *p* = 0.1834] and neither sex × diet × MB interaction [F (1, 72) = 0.5088, *p* = 0.4780] was observed regarding body weight. Additionally, even though the HFHF+MB groups decreased their weight, no statistically significant difference was observed on week 14 between the HFHF+MB vs. HFHF groups (*p* > 0.05). (Figure 2A). Furthermore, after maqui treatment, it was observed that reduction of plasmatic glucose was [main effect of time diet exposure: F (1, 72) = 32.96, *p* < 0.0001, main effect of MB: F (1, 72) = 4.855, *p* < 0.0321 and diet × MB interaction: F (1, 72) = 6.423, *p* < 0.0144] (Figure 2B); cholesterol [main effect of time diet exposure: F (1, 72) = 22.19, *p* < 0.0001, main effect of MB): F (1, 72) = 15.24, *p* < 0.0321 and diet × MB interaction: F (1, 72) = 5.939, *p* < 0.0177] (Figure 2C); and triacylglycerides [main effect of time diet exposure: F (1, 72) = 31.22, *p* < 0.0001, main effect of MB: F (1, 72) = 10.79, *p* < 0.0017 and diet × MB interaction: F (1, 72) = 11.15, *p* < 0.0014 (Figure 2D), being not statistically significantly different than the control groups. Even more, the decrease in cholesterol and triacylglycerides of the HFHF+MB groups was statistically significantly different compared to the HFHF groups (*p* < 0.05). Neither sex difference, sex × diet interaction, sex × MB interaction, nor sex × diet × MB interactions (*p* > 0.05) effect was observed in glucose, cholesterol, and triacylglycerides. With the TyG index, we observed that HFHF increases insulin resistance, while the MB treatment decreases it (Figure 1E). Finally, we observed that the HFHF diet increased systolic and diastolic blood pressure in males and females, while the administration of MB decreased these values (Table 1).

### 3.2. Effect of MB Administration on Oxidative Stress Markers in Rats Fed HFHF Diet

After a two-week MB treatment, it was observed the main effect of diet exposure [F (1, 32) = 32.09, *p* < 0.0001] with the main sex effect [F (females > males): (1, 32) = 21.46, *p* < 0.0001) and no interaction between diet and sex [F (1, 32) = 2.018, *p* < 0.1651) on MDA content. Nevertheless, even when the MB groups reported the main effect of the treatment [F (1, 32) = 23.58, *p* = 0.0001], there was no interaction between sex × MB [F (1, 32) = 0.3735, *p* = 0.5454]. However, it was observed a diet × MB interaction [F (1, 32) = 6.401, *p* = 0.0165]. No sex × diet × MB interaction was observed [F (1, 32) = 0.0114, *p* = 0.9155] about the MDA content. Interestingly, the MDA content was decreased in the HFHF+MB groups compared to the HFHF groups in both sexes (*p* < 0.05) and without a statistical difference when compared to their control groups (*p* > 0.9999) (Figure 3). Moreover, after the MB treatment, an increase in the SOD activity was observed [main effect of time diet exposure: F (1, 32) = 26.58, *p* < 0.0001, main effect of MB: F (1, 32) = 69.31, *p* < 0.0001 and diet × MB interaction: F (1, 32) = 22.25, *p* < 0.0001]. Also, it was observed a main effect of sex [F (1, 32) = 6.090, *p* < 0.0191], sex × diet interaction F (1, 32) = 6.600, *p* < 0.0151], and sex × MB interaction [F (1, 32) = 4.180, p < 0.0492]. No statistically significant difference was observed when analyzing the sex × diet × MB interaction [F (1, 32) = 3.776, *p* = 0.0608]. After the two weeks of treatment, the SOD activity was increased in the HFHF+MB groups compared to the HFHF groups in both sexes (*p* < 0.001) and without a statistical difference when compared to their control groups (*p* > 0.5500) (Figure 4). At last, it was observed the main effect of diet exposure [F (1, 33) = 5.252, *p* < 0.0284] with the main sex effect [F (males > females): (1, 33) = 4.702, *p* < 0.0374) and no interaction between diet and sex [F (1, 33) = 3.190, *p* < 0.0833) on protein carbonyl groups content. Nonetheless, even when the MB groups reported the main effect of the treatment [F (1, 33) = 25.19, *p* = 0.0001], no interaction between sex × MB [F (1, 33) = 2.052, *p* = 0.1614]. However, it was observed a diet × MB interaction [F (1, 33) = 21.32, *p* < 0.0001]. When analyzing the sex × diet × MB interaction, there was no statistically significant difference in protein carbonyl group content [F (1, 33) = 0.5508, *p* = 0.4633]. Remarkably, the protein carbonyl group content was decreased in the HFHF+MB groups compared to the HFHF groups in both sexes (*p* < 0.05) and without a statistical difference when compared to their control groups (*p* > 0.7500) (Figure 5).

### 3.3. In Silico Studies Results

From docking studies, we found that delphinidin derivatives show higher theoretical affinity than reference ligands in α-glucosidase and PPAR-γ and are very close to the theoretical affinity values of the PTP1B reference ligand (Table 2). Also, we observed that a higher number of carbohydrate units leads to an increased theoretical affinity than aglycones. Nevertheless, even when D3G5G and D3S have the same number of carbohydrates, substitution in position 5 leads to a decrease in theoretical affinity in PTP1B, PPAR-γ, and α-glucosidase. However, substitution in position 5 leads to an increase in PPAR-α and a decrease in PPAR-γ theoretical affinity (Table 3). Regarding ACE, delphinidin, and cyanidin sambubiosides showed the highest theoretical affinity, similar or even higher, to the reference ligand lisinopril. Adding sugar residues increases the hydrogen bond interactions, particularly with the sugar moiety in position 3 of delphinidin.

The ligand binding calculations (Table 4) carried out on the molecular dynamics simulations on the complexes with ACE, α-glucosidase and de PPAR receptor revealed that none of the compounds has the highest affinity against all targets: delphinidin D has the highest theoretical affinity to α-glucosidase and ACE, while D3G to PPAR-γ and D3S to PPAR-α. In the protocol implemented in Yasara, more positive ligand energy values indicate higher affinity.

## 4. Discussion

### 4.1. Effect of MB Administration on Biochemical Parameters

The bioactive compounds of MB, mainly anthocyanins, have been associated with both hypoglycemic and lipid-lowering properties. In our study, we combined in vivo plus in silico analyses to evaluate the effect of the freeze-dried MB on the parameters related to MetS in both sex-standardized animal models induced by an HFHF diet. As previously reported by our group, after 12 weeks of the HFHF diet, both sexes’ animals developed parameters of the MetS [17].

After developing the MetS model, a two-week treatment with MB was followed. When analyzing the weight, a non-statistically significant reduction in the increase of the HFHF+maqui groups was observed compared to the HFHF groups, even though the HFHF+maqui groups were still statistically significant after the maqui treatment compared to control groups.

However, there is previous evidence that MB can decrease weight. In this sense, Sandoval et al. reported that C57BL6/J mice obese mice (16 weeks of high-fat Diet (HFD)) supplemented with MB reduce weight gain. In their study, Sandoval et al. associated loss in weight gain with differential expression of de novo lipogenesis, fatty acid oxidation genes, and multilocular lipid droplet formation in white adipose tissue. The animals do not change their food consumption after MB treatment, despite a previous report by Sandoval et al. that informed hyperphagic behavior [11].

We observed that the administration of MB decreases triacylglycerides and cholesterol. This result is similar to that reported by Alvarado et al. in which the administration of Delphinol (standardized MB extract) treatment improves the blood lipid profile of prediabetic individuals in a three-month clinical trial [28].

Concerning glucose, some reports have studied the in vivo and clinical effects of MB extract on several aspects of glucose metabolism and insulin response [11,12,28,29,30]. In the present study, we observed a non-statistically reduction in the concentration of fasting blood glucose in the HFHF+MB groups compared to the HFHF groups and control groups. However, we did observe a statistically significant decrease in insulin resistance in the HFHF-MB group compared to the HFHF-MB group. Some in vivo studies have been performed in animal models of diabetes. Sandoval et al. demonstrated that MB extract supplementation increases insulin response in obese mice fed HFD [11]. Alvarado reports that Delphinidin-rich MB extract (Delphinol^®^) lowers fasting and postprandial glycemia and insulinemia in prediabetic individuals during oral glucose tolerance tests [29]. Hidalgo et al. showed long-term effects of Delphinol (standardized MB extract with ≥25% delphinidins and ≥35% total anthocyanins), reducing postprandial blood glucose and insulin in streptozotocin rats as compared to the placebo group and achieved values statically undifferentiated from non-diabetic rats [30].

As we observed, MB has beneficial effects on lipid and glucose metabolism at the clinical level and in different animal models of obesity, diabetes, and metabolic syndrome. On the other hand, some studies have highlighted the importance of natural antioxidants present in fruits to protect against metabolic diseases [31]. Therefore, the fact that MB can reduce weight, glucose, cholesterol, triacylglycerides, and insulin resistance levels suggests that MB, compared to other fruits, has a wide biological potential, being a promising target for study as a cardiovascular protector.

### 4.2. Effect of MB Administration on Oxidative Stress Markers

Numerous studies revealed that an increase in the level of reactive oxygen species (ROS) in the peripheral blood from accumulated fat is initiating insulin resistance in different adipose tissues, skeletal muscles, and other diabetic complications. After establishing oxidative stress-induced insulin resistance, it triggers the production of ROS, leading to primary oxidative reactions that cause further damage at the cellular and mitochondrial levels [32]. In our results, we observed in the serum of male and female rats that the administration of MB improved the concentration of MDA, the activity of SOD, and the formation of carbonyls in the groups subjected to a diet high in fat and fructose. One study found that levels of oxidative stress markers were decreased by 50 mg/kg MB, and the decrease in oxidative stress markers was maintained up to the 100 mg/kg MB dose [15]. In addition, some polyphenols such as epigallocatechin-3-gallate have been observed to improve insulin resistance by promoting glucose uptake and the antioxidant enzymes superoxide dismutase (SOD) and glutathione peroxidase, attenuating oxidative stress and the inflammatory markers malondialdehyde, tumor necrosis factor 1 alpha (TNF-α), interleukin 6 (IL-6), and nuclear factor kappa beta (NF-κβ) [33].

Several mechanisms have been proposed to explain the multiple bioactivities of MB components. The use of molecular docking + molecular dynamics simulation has been proven useful for the prediction or delimitation of the potential molecular mechanism of action of some molecules, particularly those of natural origin and in potential multitarget ligands [34,35,36,37,38,39,40,41].

### 4.3. In Silico Studies Conclusions and Their Correlation with In Vivo Experiments Results

The conclusions from previous molecular docking studies suggest PTP1B, α-glucosidase, DPP-IV, and PPAR-γ as plausible targets that explain the hypoglycemia activity of MB extract [42,43,44]. In these studies, aglycones and 3-glucosyl substituted derivatives are responsible for the diversity of biological actions, partially supported by experimental evidence. However, we considered that it was necessary to perform additional in silico studies to evaluate the effects of different glycosylation patterns in anthocyanins present in MB. We found that delphinidin derivatives showed higher theoretical affinity than reference ligands in α-glucosidase and PPAR-γ and very close to the theoretical affinity values of the PTP1B reference ligand, which confirms the experimental evidence that points to these biological targets to explain the glucose-lowering activity of the anthocyanins present in MB extracts [45,46]. A higher number of carbohydrate units increases theoretical affinity than aglycones. However, this can be explained by a greater possibility of molecular interactions due to the larger molecular size. Still, when comparing D3G5G and D3S, which have the same number of carbohydrate units being the main difference in these sugar units’ connectivity, it is revealed that substitution in position 5 leads to a decrease in theoretical affinity in PTP1B, PPAR-γ, and α-glucosidase. As expected, due to the high number of hydroxyl groups, these compounds interact through several hydrogen bonds, which can be observed in Appendix A.

Apropos of lipids, we observed that total cholesterol and triacylglycerides in the HFHF+maqui groups were diminished in a statistically significant manner compared to the HFHF groups. Remarkably, the reduction of these lipids was comparable to the control groups. In contrast to the present results, Alvarado et al., in the previously mentioned clinical trial, reported that chronic administration of Delphinol significantly decreases blood lipids such as low-density lipoprotein (LDL), very-low-density lipoprotein (VLDL) with an increase of HDL without changes in total cholesterol and triacylglycerides [28]. Much of the lipid-lowering effects of MB extracts can be attributed to anthocyanins’ antioxidant effect. However, cyanidin has been reported as an agonist of PPAR-α [45]. PPAR-α is the principal transcription factor of lipid metabolism in the liver. Previous reports indicate that PPAR-α enhances the uptake, utilization, and catabolism of fatty acids by gene upregulation in the transport, binding, and activation of fatty acid and β-oxidation [47]. Our docking study results also suggest PPAR-α as a potential target of MB anthocyanins. Substitution in position 5 leads to an increase in PPAR-α theoretical affinity, which is attributable to additional hydrophobic interactions with Asn 219, Cys 275, and Glu286, as seen in Appendix A. Interestingly, this substitution pattern opposes the docking results in PPAR-γ, where substitution in position 5 led to a decrease in theoretical affinity. These results suggest that there could be some synergism of MB components where some compounds have antihyperglycemic activity while others have lipid-lowering properties.

On the other hand, Ojeda et al. reported the ACE inhibitory activity of delphinidin and cyanidin sambubiosides from Hibiscus sabdariffa, which seems to explain the antihypertensive effect of the extracts [48]. This information suggests that MB could also have antihypertensive properties. In our docking study, these derivatives showed high theoretical affinity towards ACE, similar or even higher to the reference ligand lisinopril. Appendix A depicts the predicted poses for delphinidin and its derivatives within the active site of ACE. Adding sugar residues increases the hydrogen bond interactions, particularly with the sugar moiety in position 3. However, substituting an additional glucose unit in position 5 leads to a loss of π-π interactions.

From the molecular docking results and experimental data, it appears that MB components exert their positive effects on MetS factors through interaction with α-glucosidase, PPAR-α, PPAR-γ, and ACE. Although the prediction of ligand potential orientation within the binding site is helpful, more robust methods than molecular docking are needed to estimate ligand affinity more confidently. Molecular Dynamics (MD) simulations are a more rigorous tool that has helped to understand some structural features of proteins, the effects of the aqueous media, and the calculation of binding free energies, becoming a standard in drug design and development [49,50,51]. The combination of virtual screening using molecular docking on large datasets to filter the most promising ligands and molecular dynamics simulations to calculate more detailed interaction energies in the most promising compounds is frequently used [25,52,53,54].

From the ligand binding energy calculations, we found that none of the compounds has the highest affinity against all targets: delphinidin D has the highest theoretical affinity to α-glucosidase and ACE, while D3G to PPAR-γ and D3S to PPAR-α. These results reinforce the hypothesis that the multiple biological effects observed are the sum of interactions of some compounds present in the MB extract with some biotargets related to metabolic syndrome. While D and D3G could be responsible for the hypoglycemic activity due to their effect on α-glucosidase and PPAR-γ, D3S could have more impact on the lipid-lowering effect; because of its interaction with PPAR-α. Also, D is responsible for blood pressure-lowering effects due to its inhibition of ACE. Consequently, a synergistic effect of MB extract components is the most plausible explanation for the multiple biological effects observed in the in vivo study. The rigorous in silico study using the molecular docking+molecular dynamics combination strongly suggests this. Moreover, these findings correlate with some in vitro experiments that have proven the molecular mechanism of action of delphinidin and cyanidin as stated in the previous paragraphs. Finally, the in silico study allowed us to propose a general structure-activity relationship that could aid in the design and discovery of new multitarget compounds or combinations for the management of MetS. The most important finding was that the incorporation of a carbohydrate residue in position 5 has a detrimental impact on ligand affinity due to the loss of hydrogen bond interactions, while the incorporation of a saccharide unit in position 3 increases ligand affinity.

Overall, the use of the MB extract could be beneficial for MetS patients because of its multiple effects, which have been demonstrated in silico, in vitro, and in vivo animal models. However, one aspect that significantly limits these compounds’ application is their low bioavailability [54]. For this reason, the experimental results obtained in our study in a holistic model of MetS are important since they suggest that even at low systemic concentrations, anthocyanin glycosides and aglycones from MB could simultaneously act on different targets related to MetS. Therefore, MB and its compounds could be used as coadjutants in pharmacological interventions on MetS. Alvarado et al. performed two clinical studies in which metabolic parameters were evaluated after the chronic administration of Delphinol (180 mg) in subjects diagnosed with prediabetes (moderate glucose intolerance). In one, they reported a statistically significant decrease of glycosylated hemoglobin (HbA1c) after two months of consumption of Delphinol. This effect was not associated with reducing fasting insulin or glucose, nor with the Oral Glucose Tolerance Test (OGTT) without adverse effects and was well tolerated [28].

In conclusion, even though MetS is closely linked with numerous complications such as CVD and other pathological conditions, the consumption of MB can have a beneficial effect on these complications. Our findings suggest that MB has positive effects on clinical components of MetS such as weight increase, lipid (triacylglycerides and cholesterol), and glucose metabolism (fasting glucose and insulin resistance) as well as hypertension in the murine model of MetS. Also, we observed in vivo beneficial effects on the increase of oxidative stress induced by MetS. In addition, our in vitro studies indicate that the primary components of MB aqueous extract could be ligands of several metabolic targets (PTP1B, α-glucosidase, PPAR-α, PPAR-δ, PPAR-γ, CETP, ACAT-2, HMG-CoA reductase, and ACE) associated with diabetes mellitus type 2, dyslipidemia, hypertension, and obesity, as well as related to MetS parameters. All these data suggest that MB has a broad biological potential compared to other fruits, making it a promising target for the treatment of various metabolic diseases such as obesity, diabetes, and MetS.

## Figures and Tables

**Figure 1 metabolites-13-01189-f001:**
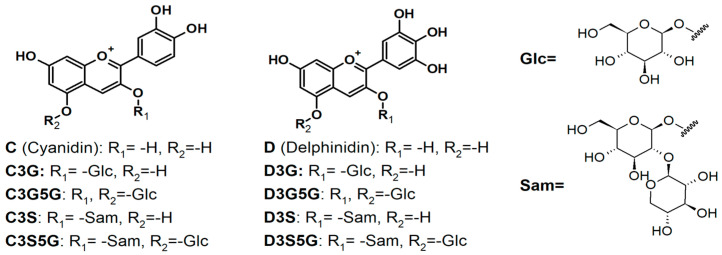
Principal anthocyanins present in MB lyophilized powder.

**Figure 2 metabolites-13-01189-f002:**
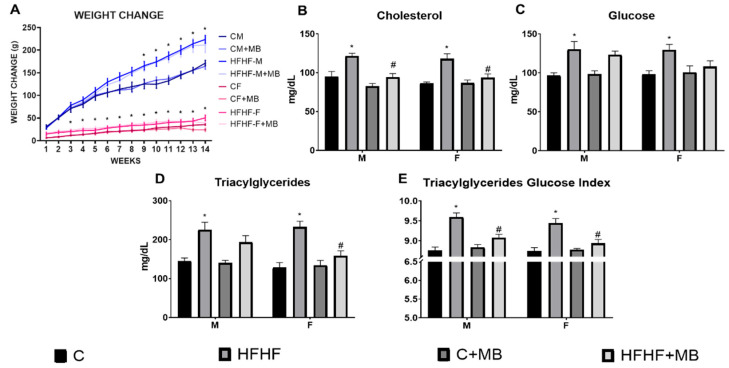
The effect after 14 days of MB administration on metabolic parameters associated with MetS on the in vivo model (**A**) Weight Change (**B**) Glucose, (**C**) Cholesterol, (**D**) Triacylglycerides, and (**E**) Triacylglicerides-glucose index. After 14 days of treatment with MB, the HFHF+MAQUI groups had a statistically significant reduction of all MetS parameters analyzed compared to the HFHF groups. Data are presented as means ± standard deviation (SD). * Represents statistically significant differences between groups compared to their own control group (HFHF groups vs. C groups) (*p* < 0.05). # Represents statistically significant differences between HFHF groups vs. HFHF+MB (*p* < 0.05).

**Figure 3 metabolites-13-01189-f003:**
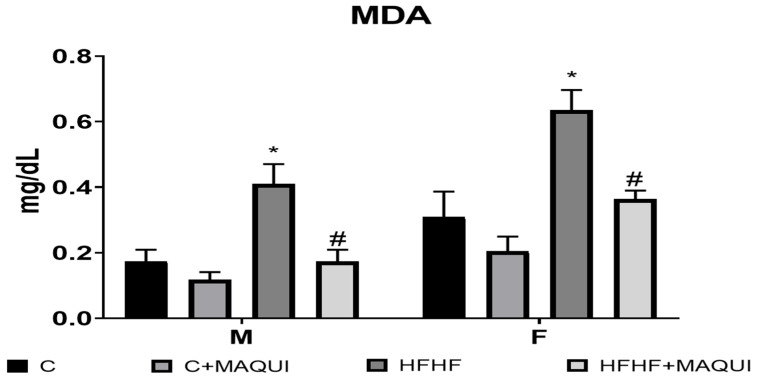
Effects of normal diets and a diet rich in fructose 30% + fat 10% on the determination of MDA in serum of male and female rats. The columns represent the mean with n = 6. The bars represent the standard error of the mean. * = significant difference with respect to the respective control. # = significant difference to the group HFHF, *p* ≤ 0.05.

**Figure 4 metabolites-13-01189-f004:**
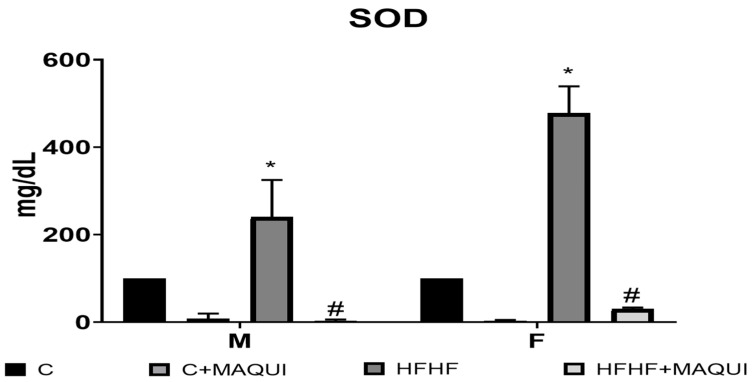
Effects of normal diets and diets rich in fructose 30% + fat 10% on the SOD activity in the serum of male and female rats; the columns represent the mean with n = 6, and the bars represent the standard error of the mean. * = significant difference with respect to the respective control. # = significant difference to the group HFHF, *p* ≤ 0.05.

**Figure 5 metabolites-13-01189-f005:**
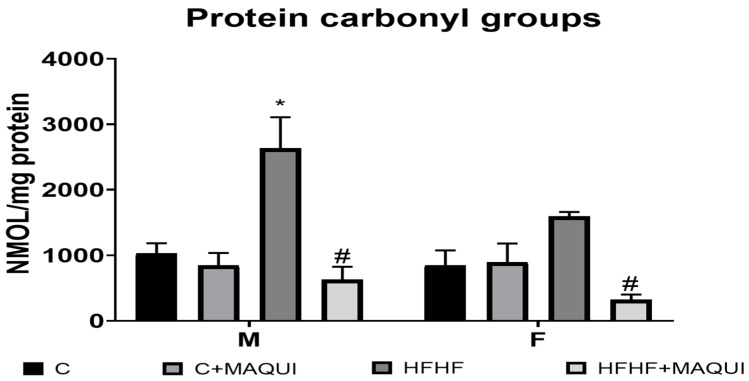
Effects of normal diets and diets rich in fructose 30% + fat 10% on the total protein carbonyls in the serum of male and female rats; the columns represent the mean with n = 6, and the bars represent the standard error of the mean. * = significant difference with respect to the respective control. # = significant difference to the group HFHF, *p* ≤ 0.05.

**Table 1 metabolites-13-01189-t001:** Effect of MB administration on systolic and diastolic pressure in rats fed HFHF diet.

Group	SAP (mmHg)	DAP (mmHg)
C-F	125.4 ± 15.46	89.0 ± 11.90
C+MB-F	119.2 ± 9.378	96.60 ± 7.884
HFHF-F	148.3 ± 14.97 *	111.0 ± 10.01 *
HFHF+MB-F	129.4 ± 8.880 #	92.8 ± 9.205 #
C-M	129.3 ± 2.562	94.3 ± 6.263
C+MB-M	121.8 ± 16.79	89.8 ± 12.73
HFHF-M	159.0 ± 6.390 *	117.25 ± 10.66 *
HFHF+MB-M	130.8 ± 12.05 #	97.8 ± 11.88 #

C: control (standard Diet), F: female, MB: maqui berry, HFHF: High fructose high-fat Diet, M: male. * Represents statistically significant differences between groups compared to their own control group (HFHF groups vs. C groups) (*p* < 0.05). # Represents statistically significant differences between HFHF groups vs. HFHF+MB (*p* < 0.05).

**Table 2 metabolites-13-01189-t002:** Docking scores calculated for anthocyanins present in maqui berry extracts in targets related to hyperglycemia.

Ligand	GPR40	DPP-IV	PPAR-δ	PTP1B	α-Glucosidase	PPAR-γ
C	−81.2	−65.0	−69.2	−86.6	−70.6	−80.7
C3G	−88.6	−78.0	−94.2	−94.0	−98.6	−112.4
C3S	−85.6	−76.3	−90.9	−108.1	−93.8	−126.0
C3G5G	−92.3	−85.8	−117.0	−104.5	−114.8	−120.4
C3S5G	−93.7	−82.5	−104.3	−102.4	−104.5	−130.5
D	−90.3	−65.4	−70.4	−86.7	−77.1	−80.6
D3G	−83.8	−85.5	−89.5	−94.8	−99.0	−107.1
D3S	−91.4	−75.4	−61.0	−101.8	−101.3	−123.5
D3G5G	−66.3	−89.0	−112.9	−88.9	−84.4	−114.5
D3S5G	−87.3	−81.1	−84.4	−102.5	−110.1	−131.1
Reference	−139.8	−102.0	−136.3	−122.2	−100.1	−107.8

**Table 3 metabolites-13-01189-t003:** Docking scores were calculated for anthocyanins present in maqui berry extracts in some targets related to dyslipidemia and hypertension.

Ligand	CETP	ACAT-2	HMG-CoA Reductase	PPAR-α	ACE
C	−69.1	−69.1	−82.8	−77.2	−77.1
C3G	−80.9	−88.6	−102.6	−107.9	−102.2
C3S	−111.6	−107.6	−101.4	−120.0	−123.2
C3G5G	−115.3	−100.7	−114.6	−135.8	−120.5
C3S5G	−109.9	−98.5	−103.0	−130.7	−118.9
D	−73.8	−72.6	−89.3	−82.2	−80.7
D3G	−90.8	−88.2	−111.0	−113.8	−115.5
D3S	−100.9	−107.9	−97.0	−126.2	−118.0
D3G5G	−98.5	−83.9	−102.7	−134.5	−116.7
D3S5G	−108.9	−108.2	−119.4	−105.6	−116.8
Reference	−111.9	−118.5	−121.4	−106.1	−118.9

**Table 4 metabolites-13-01189-t004:** Ligand binding energies (kcal/mol) calculated for delphinidin derivatives in the MD simulations.

	α-Glucosidase	PPAR-α	PPAR-γ	ACE
D	6.7	−89.8	−67.5	−123.8
D3G	−18.9	−86.3	−44.6	−211.8
D3G5G	−3.7	−176.2	−89.8	−414.5
D3S	−136.0	−46.5	−133.8	−435.5
D3S5G	−176.8	−177.2	−123.8	−571.7

## Data Availability

Data is contained within the article and Appendix A.

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
