# Peer review of "In Silico and In Vivo Evaluation of the Maqui Berry (Aristotelia chilensis (Mol.) Stuntz) on Biochemical Parameters and Oxidative Stress Markers in a Metabolic Syndrome Model"

_metabolites, 2023, doi:10.3390/metabo13121189_

Round 1
Reviewer 1 Report
Comments and Suggestions for Authors
The manuscript described the in silico and in vivo evaluation of the maqui berry (MB) on biochemical parameters and oxidative stress markers in a metabolic syndrome model. The topic is interesting. Some concerned issues should be addressed before further processing.
1.Abbreviation of a phrase should be given when the phrase was first appeared in the context, such as maqui berry (MB) in Line 34.
2.Detailed indicators of the controlled temperature and humidity conditions (Lines 98-99) should be given in Materials and Methods.
3.It is difficult to identify various treatments based on the colored lines in Figure 2 A.
4.Figure legend should be given in Figures 3, 5.
5.In the Section of Discussion, it should be focused on the reasons why your tested results were differentiated with the published reports by the others, and the mechanism leading to the effects of MB on the MetS, instead repeating the mention in the Sections of Methods and Results, such as those in Lines 380-390.
6. A list of abbreviations and their explanations should be given to the end of context.
7. Chemical composition of MB varied with growing conditions and cultivars, as well as the processing method. It is necessary to give the major composition of the tested MB sample.
Reviewer 2 Report
Comments and Suggestions for Authors
The study aims to evaluate the molecular mechanisms implied to improve the metabolic parameters of mets. It is an exciting theme. However, the introduction needs to offer a theoretical framework that justifies the accomplishment of the study. In addition, a more objective description of the objectives and hypothesis of the study needs to be included. Regarding the methodology, (1) provide more information on MB characteristics in terms of active components, such as anthocyanins and other compounds, (2) provide additional details on the high fructose diet and high fat used, including the exact composition of the diet, (3) Inform why 80 animals were used. Inform the details of the sample size calculation. The results are sufficiently detailed in the text. The graphs and tables properly complement the results described in the text. The study discussion collects information, results, and observations without a clear argument structure. Organize the discussion according to the main results and highlight the clinical or scientific implications of each result. Although the authors mentioned the results of the in silica analysis and the results of the in vivo study, the discussion needs to establish a stronger connection between these two sets of results. Be sure to provide appropriate scientific and clinical context for the results presented. For example, the authors mention reducing body weight, glucose, cholesterol, and triglycerides after treatment with machine berry. However, it is essential to discuss what these reductions mean regarding health and metabolism. Comparison with previous studies can be improved. The authors mention that the compounds of Maqui Berry have a theoretical affinity with various enzymes and receptors but do not discuss the possible mechanisms of action by which these compounds may be affecting metabolic parameters. This is fundamental to understanding the results. The conclusion can be reformulated considering the study objectives and the main results and discussion.
Reviewer 3 Report
Comments and Suggestions for Authors
Abstract reads well.
Introduction flows nicely. Introduces the topic and the gap in the literature. Please add the hypothesis and aims at the end of the introduction. It would be nicer for the reader to understand the study.
Methods are fine
Results
Please simplify the results. What are all these values in brackets besides p-value.? Please indicate what is what in the beginning for clarity.
Please add statistics table to show main effects and interaction effects in the supplementary file. So that the results section contains only main effects and significant interaction effects if any. The sentences are very interruptive with all those numbers and p-values
Please show liver triglyceride and plasma insulin levels. Important for metabolic syndrome study.
Discussion
Line 399-404 how is this paragraph related to this study?
Line 417-418 is this correct? Is it not the other way around? Oxidative stress leading to insulin resistance.
Line 423-424- did you mean MB rather than BM?
Line 428- it should be Tumor necrosis factor 1 alpha.
The docking study is ok but there is no experimental evidence to show the findings of in silico study. Most of it is hypothetical. Can you show some experimental evidence to support the in silico findings?
Where is the conclusion?
Round 2
Reviewer 2 Report
Comments and Suggestions for Authors
I would like to inform that the authors have satisfactorily made alterations to the text, addressing the suggestions and considerations provided. When it wasn't feasible to amend specific parts of the manuscript, the authors provided appropriate justifications.
The actual version is now ready for publication.
Reviewer 3 Report
Comments and Suggestions for Authors
Authors have revised the manuscript as per most of the suggestions. Manuscript has been sufficiently revised.